# Effect of Membrane Surface Area on Solute Removal Performance of Dialyzers with Fouling

**DOI:** 10.3390/membranes12070684

**Published:** 2022-07-01

**Authors:** Takayoshi Kiguchi, Hiromi Ito, Akihiro C. Yamashita

**Affiliations:** 1Department of Chemical Science and Technology, Faculty of Bioscience and Applied Chemistry, Hosei University, 3-7-2, Kajino-cho, Koganei 184-8584, Tokyo, Japan; takayoshi.kiguchi.38@hosei.ac.jp; 2Department of Applied Chemistry, Graduate School of Science and Engineering, Hosei University, 3-7-2, Kajino-cho, Koganei 184-8584, Tokyo, Japan; hrmit.07141311@gmail.com

**Keywords:** dialysis, membrane fouling, membrane surface area, internal filtration, packing density of hollow fiber

## Abstract

In a clinical situation, since membrane fouling often causes the reduction of solute removal performance of the dialyzer, it is necessary to evaluate the performance of the dialyzer, considering the effects of fouling even in aqueous in vitro experiments that are useful for the better design of dialyzers. We replicated the membrane fouling by immobilizing albumin on the membrane in a dialyzer using glutaraldehyde as a stabilizer. The modules of various membrane surface areas with and without replication of the fouling were used for performance evaluation of solute (creatinine, vitamin B_12_, and inulin) removal in dialysis experiments in vitro. Clearances for these solutes in the modules with fouling were lower than those without fouling. Furthermore, the smaller the surface area, the larger the fouling effect was observed in all solutes. Calculated pressure distribution in a module by using a mathematical model showed that the solute removal performance might be greatly affected by the rate of internal filtration that enhances the solute removal, especially for larger solutes. The increase in the rate of internal filtration should contribute to improving the solute removal performance of the dialyzer, with a higher effect in modules with a larger membrane surface area.

## 1. Introduction

Dialysis is the most popular treatment for patients with end-stage kidney disease that aims to remove excess fluid and uremic toxins accumulated in the patient body. For the past few decades, many kinds of dialyzers have been developed, and the performance of dialyzers has been improved to achieve more effective solute removal. The high-flux dialyzer in which the internal filtration is enhanced with a membrane of high hydraulic permeability shows high solute removal performance due to the contribution of a considerable amount of convection across the membrane in addition to diffusion. Since the internal filtration rate is affected by a pressure drop of blood and dialysate in the dialyzer as well as the hydraulic permeability of the membrane, adequate design of dialyzers, i.e., the optimization of many factors such as effective length [1], the inner diameter of the hollow fiber [2], packing density of the hollow fiber [3,4], the inner diameter of the housing (outer casing) [5] and the membrane surface area [6,7] are important. Ronco et al. measured the pressure drop and calculated the internal filtration rate in their works using labeled albumin as a marker [2,8]. Sakiyama et al. experimentally measured the internal filtration rate of high-flux dialyzers using Doppler ultrasonography, indicating that at most, about 58 mL/min of internal filtration occurred at a blood flow rate of 350 mL/min [9]. In addition to these, Fukuda et al. investigated the effect of the baffle structure and taper length on the performance of a dialyzer by evaluating the clearance of urea, creatinine, and β_2_-microgroblin [10]. Nakashima et al. visualized dialysate flow by computed tomography in two kinds of the dialyzer for a better design of the dialyzer [11]. Thus, the findings can be obtained from in vitro laboratory experiments that are useful for investigating the effects of various factors on the performance of the dialyzer [12]. We also have evaluated the solute removal performance and biocompatibility of various dialyzers in vitro [13,14,15,16,17]. On the other hand, plasma proteins are adsorbed on the membrane during dialysis treatment, resulting in reduced solute removal performance of the dialyzer [18,19,20,21,22,23,24]. Therefore, it is necessary to consider the effects of membrane fouling on the solute removal performance of dialyzers. In order to evaluate the performance of dialyzers with fouling in the laboratory, immobilization of proteins on membranes using glutaraldehyde as a stabilizer is applicable to the replication of fouling [25,26,27]. For example, Yamamoto et al. immobilized adhering proteins to the membrane with glutaraldehyde after dialysis experiments using bovine blood and evaluated the degree of fouling at each position of the dialyzer [28]. In our previous work, we have investigated a method of replicating the fouling in aqueous experiments under various immobilization conditions by using albumin as a foulant [29].

In the present work, we aimed to investigate the effect of membrane surface area on solute removal performance in aqueous in vitro experiments, especially in dialyzers with protein fouling. For this investigation, we immobilized albumin on the membrane in the same series of dialyzers with different membrane surface areas by our replicating method and evaluated the clearance and the overall mass transfer coefficient of the following three solutes with different molecular weights (MW [-]); creatinine (MW 113), vitamin B_12_ (MW 1355) and inulin (MW 5000).

## 2. Experimental

### 2.1. Materials

A commercial dialyzer series of APS-SA (polysulfone, PSf; Asahi Kasei Medical Co., Ltd., Tokyo, Japan) and that of MFX-S eco (polyethersulfone, PES; Nipro Co., Osaka, Japan) with different membrane surface areas were investigated in this study. The inner and outer diameters of the hollow fiber in APS-SA were measured by digital microscope (VH-7000; KEYENCE Co., Osaka, Japan).

In order to evaluate the solute permeability of the dialyzer, creatinine (Mw 113; Fujifilm Wako Pure Chemical Co., Osaka, Japan), vitamin B_12_ (Mw 1355; Fujifilm Wako Pure Chemical Co.), and inulin (MW 5000; Alfa Aesar Thermo Scientific Chemicals Inc., Waltham, MA, USA) were used as test solutes. Albumin (Fujifilm Wako Pure Chemical Co.; Mw 66,000) and 25 wt% glutaraldehyde solution (Fujifilm Wako Pure Chemical Co.) were used for the replication of fouling on the membrane. In order to prepare phosphate buffer solution of pH 7.4, potassium dihydrogen phosphate (Fujifilm Wako Pure Chemical Co.) and sodium hydroxide (Kanto Chemical Co., Inc., Tokyo, Japan) were used. In order to measure the concentration of inulin by UV-Visible spectrophotometer (UV-1280, Shimadzu Co., Kyoto, Japan), a method using 3-indoleacetic acid, hydrochloric acid, and polyoxyethylene lauryl ether (all three were purchased from Fujifilm Wako Pure Chemical Co.) was employed.

### 2.2. Albumin Immobilization

Albumin immobilization to hollow fiber membranes was performed in the following two steps. In the first step, 6.8 g of potassium dihydrogen phosphate was completely dissolved in ion-exchanged water by using a magnetic stirrer. Sodium hydroxide solution prepared to a concentration of 2.0 M was added to the solution until the pH of solution reached 7.4. After adjusting pH, 1.0 g of albumin was dissolved in the solution, and ion-exchanged water was added to prepare 500 mL of albumin solution (albumin concentration was 2.0 g/L). A test circuit for circulating the albumin solution into the test module (dialyzers/diafilters) was assembled, as shown in Figure 1a, and the test model was washed with ion-exchanged water for 10 min. The albumin solution was stirred and maintained at thirty-seven degrees Celsius by a hot stirrer and was pumped into a dialyzer at *Q*_Alb_ = 200 mL/min using a blood pump (DKP-01; Nikkiso Co., Ltd., Tokyo, Japan). During the circulation of the albumin solution, the further port for filtrate close to the outlet of the test solution was sealed, and filtration was made at *Q*_F_ = 100 mL/min from the other port near the inlet of the test solution using a filtration pump (PRS-12; Nikkiso Co., Ltd.). The circulation was made for one hour. After the circulation was over, the residual albumin concentration in the test solution was measured by UV-Visible spectrophotometer to calculate the adsorbed amount of albumin by the test module.

In the second step, 27.2 g of potassium dihydrogen phosphate was added to ion-exchanged water, and 2.0 M of sodium hydroxide solution was added to the solution until the pH of solution reached 7.4, stirring the solution with a magnetic stirrer. Ion-exchanged water was added to the solution to be a total volume of 1840 mL, and the solution was mixed with 160 mL of 25% glutaraldehyde (GA) solution. The test circuit to supply GA solution to the test module for albumin immobilization was constructed as shown in Figure 1b. The GA solution was stirred and maintained at thirty-seven degrees Celsius by a hot stirrer and was pumped into the test module at *Q*_GA_ = 200 mL/min using a blood pump. While the GA solution was supplied to the test module, the outlet port of the test solution was sealed, and the filtrate was drained from two filtration ports. After 2000 mL of the GA solution was supplied, the test module was removed from the circuit and was kept in a refrigerator at four degrees Celsius for 20 h.

### 2.3. Dialysis Experiment

Figure 1c shows the test circuit for the measurement of solute clearances. Before the test module was assembled in the circuit, ion-exchanged water was circulated in the circuit for 20 min for washing. After setting the test module, the test solution, as well as dialysate in the test module, was carefully deaerated. Then, ion-exchanged water was pumped into the test module for 10 min for washing.

Ten liters of ion-exchanged water and 0.30 g of test solute were mixed and agitated with a turbine-type stirrer. The test solution was pumped into the test module at *Q*_B_ = 200 mL/min. Ion-exchanged water was used as dialysate that was operated in the counter-current fashion at *Q*_D_ = 350, 500, or 650 mL/min. During the dialysis, the test solution was kept at thirty-seven degrees Celsius. Samples were taken at the inlet and outlet of the test module as well as at the outlet of dialysate. The concentration of these samples was measured by UV-Visible spectrophotometer at the wavelength of 234 nm for creatinine, 360 nm for vitamin B_12_ directly, and at 520 nm for inulin after coloring the solution using coloring agents [30].

### 2.4. Evaluation of Mass Transfer in Dialysis Experiments

The amount of albumin adsorbed by test module (*M*_ads_) was calculated from the following equation:(1)$$M_{\mathrm{ads}}=C_{\mathrm{ini}}\times V_{\mathrm{ini}}-C_{\mathrm{res}}\times \left(V_{\mathrm{ini}}+V_{\mathrm{pri}}\right)$$
where *C*_ini_ is the initial albumin concentration (= 2.0 g/L), *C*_res_ is the residual albumin concentration after adsorption, *V*_ini_ is the prepared volume of albumin solution (= 0.50 L), and *V*_pri_ is the priming volume of the test module and circuit, respectively.

Solute removal performance of the test module was evaluated by the solute clearance (*C*_L_) defined in the following manner:(2)$$C_{L}=\frac{C_{\mathrm{Bi}}-C_{\mathrm{Bo}}}{C_{\mathrm{Bi}}}\times Q_{B}$$
where *C*_Bi_ and *C*_Bo_ are the concentrations in the test solution at the inlet and outlet of the test module, respectively.

The reduction rate of clearance (*R*_CL_) after the membrane fouling (albumin immobilization) was defined by the following equation:(3)$$R_{\mathrm{CL}}=\left(1-\frac{C_{L\left(\mathrm{fouled}\right)}}{C_{L\left(\mathrm{un}-\mathrm{fouled}\right)}}\right)\times 100$$
where *C*_L(un-fouled)_ and *C*_L(fouled)_ are the clearance before and after immobilization of albumin, respectively.

From the engineering point of view, the overall mass transfer-area coefficient (*K*_O_*A*), a product of the overall mass transfer coefficient *K*_O_ and the effective membrane surface area *A*, was calculated from the following equation:(4)$$K_{O}A=\frac{1}{\frac{1}{Q_{B}}-\frac{1}{Q_{D}}}\times \ln\left(\frac{1-\frac{C_{L}}{Q_{D}}}{1-\frac{C_{L}}{Q_{B}}}\right)$$

Moreover, the value of *K*_O_*A* was divided by the arithmetic surface area of the dialyzer *A*_0_, *K*_O_*A*/*A*_0_, to compare the solute removal performance of the test modules with various membrane surface area.

Michaels showed the following dimensionless relationship for evaluation of dialyzers [31], i.e.,
(5)$$E=\frac{C_{L}}{Q_{B}}=\frac{1-\exp\left\{N_{T}\left(1-Z\right)\right\}}{Z-\exp\left\{N_{T}\left(1-Z\right)\right\}}$$
(6)$$Z=\frac{Q_{B}}{Q_{D}}$$
(7)$$N_{T}=\frac{K_{O}A}{Q_{B}}$$
where *E* is the extraction ratio of the solute of interest, *z* is the ratio of *Q*_B_ to *Q*_D_ and *N*_T_ is the number of mass transfer units. Consequently, the overall efficiency of the mass transfer in dialyzer in the absence of ultrafiltration is determined by *Q*_B_, *Q*_D_, and *K*_O_*A*.

In this study, the solute clearance normalized to a membrane surface area of 15,000 cm^2^ (= *C*_L15000_) was evaluated in accordance with the Japanese guideline for evaluation of solute removal performance [32], in which the performance evaluation must be made for a commercial model of 15,000 cm^2^ with *Q*_B_ = 200 mL/min and *Q*_D_ = 500 mL/min. Then values of *K*_O_*A* calculated for each specific module with various *A*_0_ were converted to the *K*_O_*A* with a membrane area of 15,000 cm^2^ (= *K*_O_*A*_15000_) using the following Equation (8), i.e.,
(8)$$\frac{K_{O}A}{A_{0}}\times 15,000=K_{O}A_{15000}$$

Then *C*_L15000_ was calculated from the following two equations with *Q*_B_ = 200 mL/min and *Q*_D_ = 500 mL/min, i.e.,
(9)$$N_{T15000}=\frac{K_{O}A_{15000}}{Q_{B}}$$
(10)$$E_{15000}=\frac{C_{L15000}}{Q_{B}}=\frac{1-\exp\left\{N_{T15000}\left(1-z\right)\right\}}{z-\exp\left\{N_{T15000}\left(1-z\right)\right\}}$$

### 2.5. Computation of Pressure Distribution in a Dialyzer

The following basic equation was used to analyze the pressure distribution in the test module [3,24,33]:(11)$$\frac{dP_{B}\left(z\right)}{dz}=-\frac{128\mu_{B}\left(z\right)}{\pi d^{4}n}Q_{B}\left(z\right)$$
(12)$$\frac{dP_{D}\left(z\right)}{dz}=\frac{32\mu_{D}\left(z\right)}{d_{e}^{2}S_{D}}Q_{D}\left(z\right)$$
(13)$$\frac{dQ_{B}\left(z\right)}{dz}=-\frac{dQ_{D}\left(z\right)}{dz}=-L_{p}\left(\frac{A}{L}\right)\left\{P_{B}\left(z\right)-P_{D}\left(z\right)\right\}$$
where *P*_B_(*z*) and *P*_D_(*z*) are the local blood, and dialysate side pressures, and *Q*_B_(*z*) and *Q*_D_(*z*) are the local blood and dialysate flow rates that vary in the longitudinal direction of the device *z*, *μ*_B_(*z*) and *μ*_D_(*z*) are the local blood and dialysate viscosities, *d* is an inner diameter of the hollow fiber, *L* is the effective length of the hollow fiber, and *L*_P_ is the hydraulic permeability of the membrane. In this paper, *μ*_B_(*z*) and *μ*_D_(*z*) were set equal to constants and were approximated to that of pure water. The number of hollow fibers *n* can be calculated in the following manner:(14)$$n=\frac{A_{0}}{\pi dL}$$

The equivalent diameter of the hollow fiber *d*_e_ and the cross-sectional area of the dialysate flow *S*_D_ can be calculated by the following equations:(15)$$d_{e}=\frac{{\left(dd\right)}^{2}-{\left(d+2\Delta x\right)}^{2}\times n}{dd-\left(d+2\Delta x\right)\times n}$$
(16)$$S_{D}=\frac{\pi }{4}\left\{{\left(dd\right)}^{2}-{\left(d+2\Delta x\right)}^{2}\times n\right\}$$
where Δ*x* is the physical thickness of the hollow fiber membrane, and *dd* is the inner diameter of the case. The hydraulic permeability of the membrane *L*_P_ and the transmembrane pressure (*TMP*) were calculated by the following equations:(17)$$L_{P}=\frac{Q_{F}}{TMP\times A_{0}}$$
(18)$$TMP=\frac{P_{\mathrm{Bi}}+P_{\mathrm{Bo}}}{2}-\frac{P_{\mathrm{Di}}+P_{\mathrm{Do}}}{2}$$
where *Q*_F_ [mL/min] is the ultrafiltration rate, *P*_Bi_ and *P*_Bo_ are the test solution pressures at the inlet and the outlet of the test solution, *P*_Di_ and *P*_Do_ are the dialysate pressures at the inlet and the outlet of the dialysate, respectively. In this study, *TMP* and *L*_P_ were calculated from the measurement of the pressure at each point at *Q*_B_ = 200 mL/min and *Q*_F_ = 100 mL/min using the test circuit shown in Figure 1c.

The packing density of the hollow fiber (PDF (%); ratio of the cross-sectional area of hollow fibers to that of the superficial) was calculated by the following equation:(19)$$\mathrm{PDF}=\frac{n\times \frac{\pi {\left(d+2\Delta x\right)}^{2}}{4}}{\frac{\pi {\left(dd\right)}^{2}}{4}}\times 100=n{\left(\frac{d+2\Delta x}{dd}\right)}^{2}\times 100$$

## 3. Results and Discussion

### 3.1. Albumin Adsorption to the Test Module for Replication of Fouling

Figure 2a shows the amount of albumin adsorbed by test modules with various membrane surface areas, *A*_0_. In both APS-SA and MFX-S eco series, the amount of albumin adsorbed on the membrane increased with an increase in *A*_0_. Comparing the results with the APS-SA series and those with the MFX-S eco series, it was found that the latter was approximately 20% higher than those of the former. MFX-S eco series are super high-flux models specifically designed for hemodiafiltration. However, the APS-SA series are also super high-flux but are dialyzers rather than diafilters, allowing a small amount of albumin leakage in a clinical situation. Then MFX-S eco series has a chance to capture more albumin inside the membrane; however, even with the largest membrane area, the adsorbed amount of albumin was only about 200 mg at most, which was only 20% of the total albumin contained in the solution. Hence, the albumin solution for immobilization contained a sufficiently large amount of albumin for all investigated models.

The amount of albumin adsorbed by the membrane was divided by *A*_0_ is shown in Figure 2b. Although values tended to decrease with the increase in *A*_0_ in both series, the rate of decrease was small, and *M*_ads_/*A*_0_ decreased by only 21.2% for APS-SA and 19.3% for MFX-S eco with the increase in *A*_0_ from 11,000 to 25,000 cm^2^. Since replicating the protein fouling was successfully performed on all investigated models regardless of *A*_0_, it may be possible to evaluate the solute removal performance of various dialyzers with fouling.

### 3.2. Evaluation of Solute Removal Performance in Two Types of Dialyzers

Figure 3 shows the clearances for creatinine with and without albumin immobilization. Figure 3a shows the relationship between *A*_0_ and creatinine clearance in the APS-SA series. In un-fouled models, the clearance increased with the increase in *A*_0_ in the low *A*_0_ range, and the clearance reached almost a plateau in the *A*_0_ greater than 15,000 cm^2^. This suggests that under *Q*_B_ = 200 mL/min, *C*_L_ = *Q*_B_ was almost achieved when *A*_0_ = 15,000 cm^2^, and increasing *A*_0_ has little effect on increasing clearance for small solutes such as creatinine. In addition, although the clearances for small solutes mainly depend on diffusion, the increase in dialysate showed a limited effect on increasing clearance because, under *Q*_B_ = 200 mL/min, most solutes are effectively removed by diffusion even with a reduced *Q*_D_ in series of these high-flux models. In the models with fouling, the clearance was significantly increased with the increase in *A*_0_ in the range of *A*_0_ < 15,000 cm^2^. Unlike un-fouled models, since the fouling effect in small models are larger, clearance showed a much greater increase with the increase in *A*_0_. However, in the range of *A*_0_ > 15,000 cm^2^, the clearance decreased slightly and showed local minima at around *A*_0_ = 21,000 cm^2^. Figure 4 shows the relationship between the packing density of the hollow fiber (PDF) and *A*_0_; APS-SA took the local minima at *A*_0_ = 15,000 cm^2^. Usually, the higher the PDF, the higher the rate of internal filtration may be expected; however, if the PDF is too high, unexpected channeling in the dialysate may cause a lower rate of mass transfer, which explains lower performance in APS-SA at *A*_0_ = 18,000 and 21,000 cm^2^.

Clearances in un-fouled MFX-S eco increased with the increase in *A*_0_ and reached a plateau in Figure 3b, showing the similar manner observed in the APS-SA series. Even with fouling models, the clearance increased and approached *Q*_B_ (= 200 mL/min) asymptotically with the increase in *A*_0_ under a little influence of the values of *Q*_D_. 

The magnitude of the effect of fouling due to the difference in membrane surface area can be found in Figure 3c,d, which show the reduction rate of clearance as a function of *A*_0_. These figures clearly tell that models with larger surface areas are less affected by the fouling in terms of removal performance for creatinine. Then models with a larger surface area can remove small solutes of interest more efficiently, even in the clinical situations with fouling.

### 3.3. Computation of Pressure Distribution in Test Models

The technical specifications of both the APS-SA and MFX-S eco series are shown in Table 1 and Table 2. APS-SA series has a relatively smaller inner diameter of the hollow fiber and are designed to increase the inner diameter of the case only for models with larger surface area. Therefore, the PDF also depends on the surface area of a dialyzer. By contrast, the MFX-S eco series are designed to retain PDF by changing both the inner diameter and the effective length of the case (Figure 4). In contrast, the PDF showed little change in the MFX-S eco series, that in the APS-SA series intricately changed with the increase in *A*_0,_ especially in the range of *A*_0_ = 13,000–21,000 cm^2^.

Figure 5 shows the theoretical calculation of the pressure distribution in the APS-SA series under *P*_Bi_ = 10 kPa (=75 mmHg), *Q*_B_ = 200 mL/min, and *Q*_D_ = 500 mL/min with no net ultrafiltration. Since the calculation was performed assuming that the test model lies on its side, the horizontal axis shows an arbitrary position *z* from the blood entrance. Accordingly, the left side of the figure (*z* = 0 cm) is the blood inlet (dialysate outlet), and the right side of the figure (*z* = 27 cm) is the dialysate inlet (blood outlet). In the z range of 0 to 13.5 cm, the pressure of the test solution is higher than that of dialysate, and normal filtration occurs from the test solution compartment to the dialysate. On the other hand, in the *z* range of 13.5 to 27 cm, since the pressure of the dialysate is higher, so-called back filtration should be expected to occur from the dialysate compartment to the test solution.

In Figure 5a–d, as *A*_0_ increases, the slope of pressure in the test solution decreases because of the increase in the number of hollow fibers, as shown by Equation (11). Moreover, the pressure drop on the dialysate side depends on the PDF. Models with *A*_0_ = 8000 and 13,000 cm^2^ showed the dialysate pressure drop of approximately 2.7 and 3.1 kPa, respectively. On the other hand, a model with *A*_0_ = 15,000 cm^2^, having relatively lower PDF, showed a dialysate pressure drop of approximately 1.6 kPa. From these calculations, it can be found that the PDF greatly affects the dialysate pressure drop in the APS-SA series.

Figure 6 shows the relationship between the internal filtration flow rate calculated from pressure distribution and *A*_0_, depicted by smooth curves. In the APS-SA series, the rate of internal filtration was relatively low because of relatively low hydraulic permeability and did not increase much with the increase in *A*_0_. Moreover, the change looked parallel to that found with PDF in Figure 4 [34]. At a model with *A*_0_ = 15,000 cm^2^, the low PDF of the module led to the low dialysate pressure drop as calculated in Figure 5, resulting in the decrease in *Q*_IF_. On the other hand, in the MFX-S eco series, the rate of internal filtration was relatively high because of relatively high hydraulic permeability and increased monotonously and greatly with the increase in *A*_0_ since the PDF of the MFX-S eco series was almost constantly high at around 60% (Figure 4). Since approximately 50 mL/min of the internal filtration flow rate may be expected in the MFX-S eco series for *A*_0_ > 21,000 cm^2^, the clearance would not decrease so much even under the fouling is formed.

### 3.4. Effect of Solute Molecular Weight on the Removal Performance of Dialyzer

Clearances for vitamin B_12_ and inulin in the APS-SA series are shown as a function of *A*_0_ in Figure 7a and 7b, respectively. The clearance for vitamin B_12_ in un-fouled models monotonously increased with the increase in *A*_0_ with a little asymptotic effect for *A*_0_ > 18,000 cm^2^ because the clearance approached *Q*_B_, a similar effect of which was observed for creatinine in Figure 3. In Figure 7b, the clearance for inulin also monotonously increased with the increase in *A*_0_, the slope of which was almost the same as that found in vitamin B_12_. This fact tells that models with larger surface areas can more effectively remove middle molecules such as inulin.

The overall mass transfer-area coefficient divided by *A*_0_ (*K*_O_*A*/*A*_0_) for creatinine, vitamin B_12_, and inulin are shown in Figure 8a, 8b, and 8c, respectively, as a function of *A*_0_ in the APS-SA series in the same scale for comparison. For creatinine, *K*_O_*A*/*A*_0_ decreased with the increase in *A*_0_ in the un-fouled dialyzers, although under fouling, *K*_O_*A*/*A*_0_ did not show much decrease with *A*_0_. The former effect may be explained by the fact that *C*_L_ is approaching *Q*_B,_ which must be met for all models with various surface areas. For vitamin B_12_ and inulin, values of *K*_O_*A*/*A*_0_ decreased with a much lower slope with the increase in *A*_0_ in un-fouled models, and those were almost constant under the fouling.

Clearances normalized to the effective surface area of 15,000 cm^2^ (*C*_L15000_) as a function of *A*_0_ for creatinine, vitamin B_12_, and inulin are shown in Figure 9a, 9b, and 9c, respectively. Values of *C*_L15000_ for creatinine in the range *A*_0_ > 15,000 cm^2^ were lower than the value at *A*_0_ = 15,000 cm^2^, which clearly shows that it is not recommended to estimate the clearance at *A*_0_ = 15,000 cm^2^ by using larger models since the clearance with *A*_0_ = 15,000 cm^2^ is used as standard representing the series of the same brand. On the other hand, values of *C*_L15000_ for vitamin B_12_ and inulin did not show a particular decrease with the increase in *A*_0_ under fouling. This is partly because the increase in the internal filtration flow rate in the large membrane surface area contributed to an increase in the solute removal performance. This effect has a greater impact on vitamin B_12_ than on creatinine, which can sufficiently be removed by diffusion; however, the effect has a little smaller impact on inulin, whose sieving coefficient is significantly smaller than unity. Therefore, the improvement of solute removal by internal filtration may be dependent on the molecular size of the solute of interest as well as membrane permeability.

## 4. Conclusions

By immobilizing albumin on the membrane using glutaraldehyde as a stabilizer, replicating the fouling was successfully performed on various modules with different membrane surface areas. The modules of various membrane surface areas with and without replication of the fouling were used for the performance evaluation of solute removal.

When compared to modules without fouling, creatinine clearance was lower in the modules with fouling; moreover, the smaller the surface area, the larger the fouling effect was observed. In addition, in the APS-SA series with fouling, the clearance decreased slightly with the increase in *A*_0_ in the range of *A*_0_ > 15,000 cm^2^ and showed local minima around *A*_0_ = 21,000 cm^2^. It was suggested that unexpected channeling in dialysate due to the high packing density of the hollow fiber may have decreased the mass transfer of creatinine.

The computation of pressure distribution in modules showed that the change in internal filtration flow rate seemed parallel to that found in the packing density of the hollow fiber in the APS-SA series. By contrast, the internal filtration flow rate monotonously increased with the increase in membrane surface area in the MFX-S eco series. The design of the module significantly affected the pressure drop in the module and caused a change in the internal filtration flow rate and the solute removal performance.

The solute removal performances of creatinine, vitamin B_12,_ and inulin in the APS-SA series were evaluated by using the clearance normalized to the effective surface area of 15,000 cm^2^ (*C*_L15000_). Although the normalized clearance of creatinine decreased with the increase in *A*_0_ in the range *A*_0_ > 15,000 cm^2^, a particular reduction in vitamin B_12_ and inulin clearance was not shown in the modules with fouling. Thus, the results found that the increase in internal filtration flow rate in larger membrane surface area contributed to improving the solute removal performance in modules whose performance decreased by fouling and that the effect may be dependent on the molecular size of the solute.

## Figures and Tables

**Figure 1 membranes-12-00684-f001:**
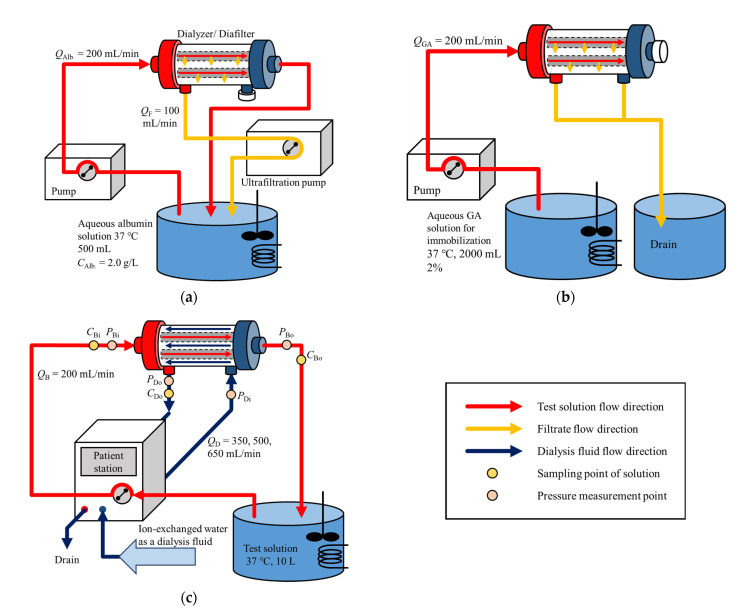
Test circuit for (**a**) albumin adsorption, (**b**) immobilization by glutaraldehyde, and (**c**) measurement of solute clearances.

**Figure 2 membranes-12-00684-f002:**
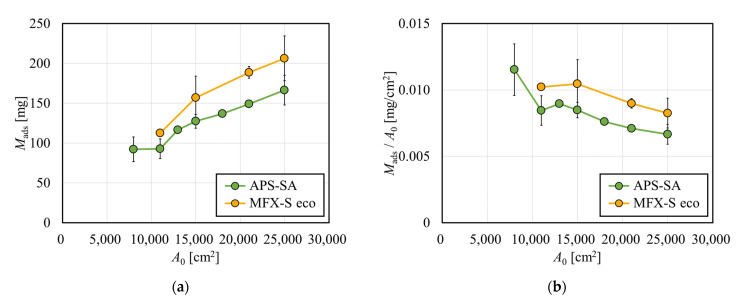
Effect of the arithmetic surface area (*A*_0_) on (**a**) the amount of albumin adsorbed by test modules (*M*_ads_) and (**b**) the amount of albumin adsorbed per unit surface area of albumin (n = 3).

**Figure 3 membranes-12-00684-f003:**
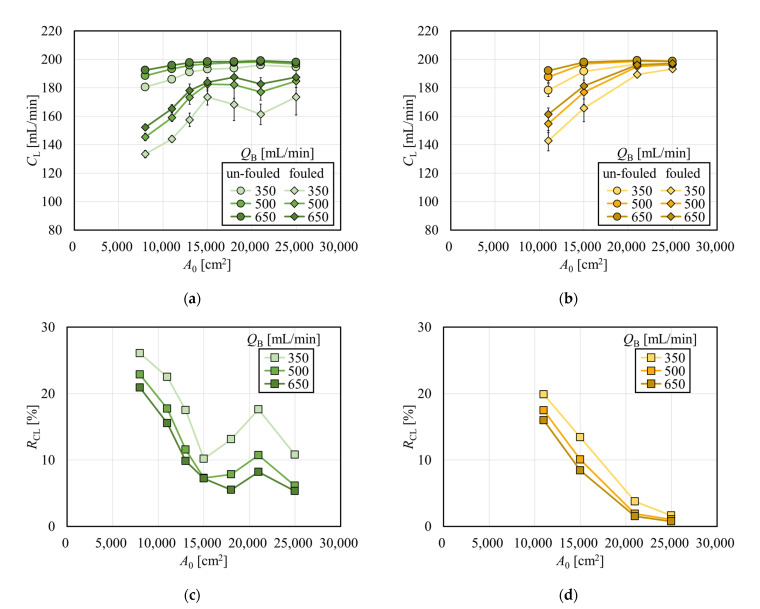
Effect of membrane fouling on the solute removal performance. (**a**) Creatinine clearances in APS-SA series with and without immobilization of albumin (*Q*_D_ = 500 mL/min, n = 3). (**b**) Creatinine clearances in MFS-S eco series with and without immobilization of albumin (*Q*_D_ = 500 mL/min, n = 3). (**c**) Reduction rate of clearance in APS-SA series. (**d**) Reduction rate of clearance in MFX-S eco series.

**Figure 4 membranes-12-00684-f004:**
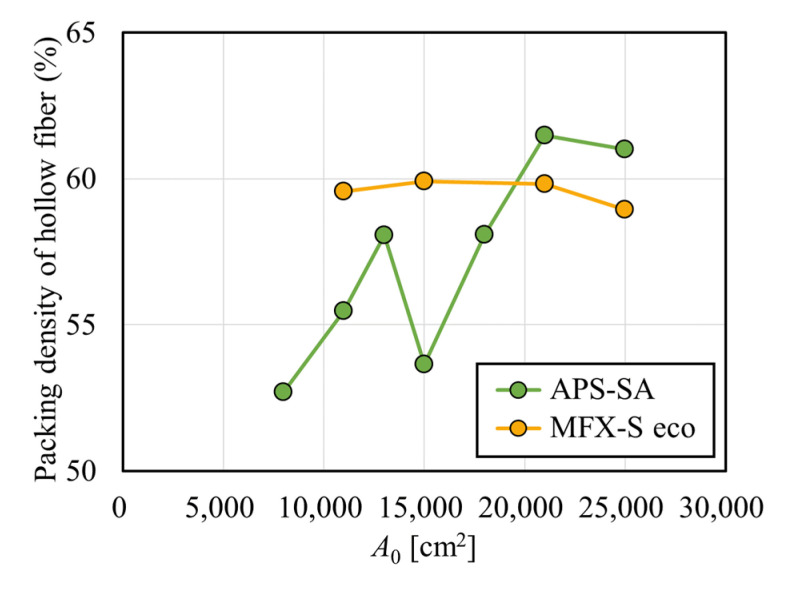
Relationship between *A*_0_ and the packing density of the hollow fiber (PDF).

**Figure 5 membranes-12-00684-f005:**
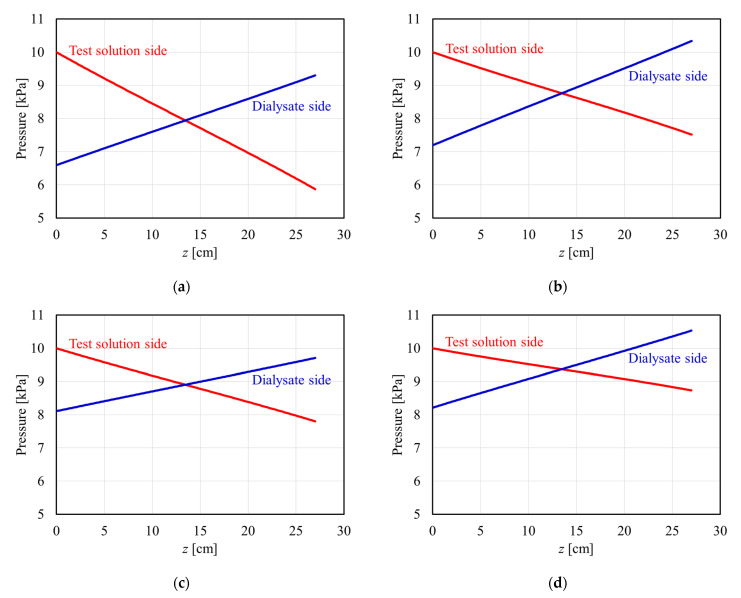
Pressure distribution in APS-SA series with a membrane surface area of (**a**) 8000 cm^2^, (**b**) 13,000 cm^2^, (**c**) 15,000 cm^2^, and (**d**) 25,000 cm^2^.

**Figure 6 membranes-12-00684-f006:**
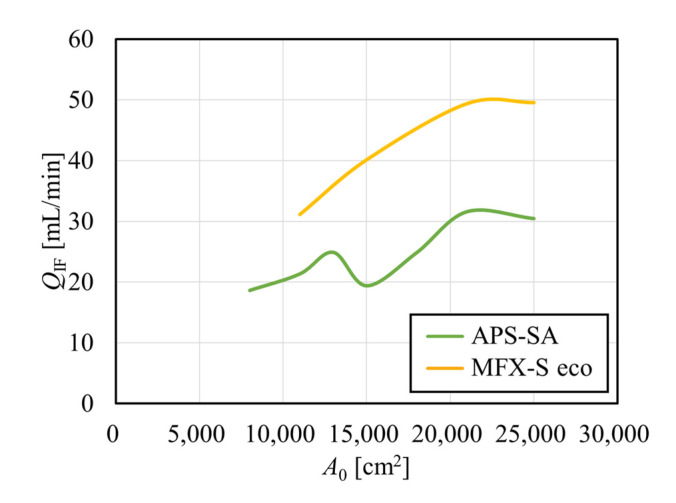
Relationship between the arithmetic surface area and the internal filtration flow rate calculated from pressure distribution.

**Figure 7 membranes-12-00684-f007:**
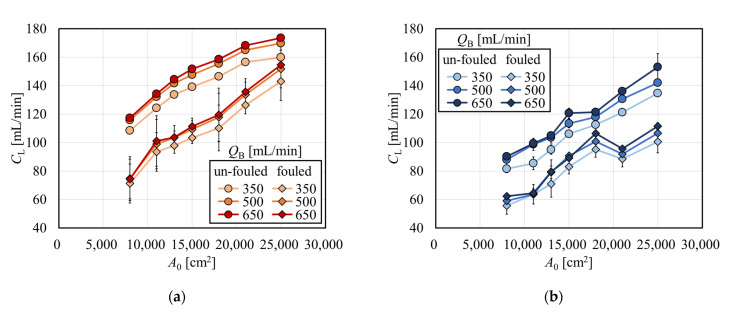
Clearance for (**a**) vitamin B_12_ and (**b**) inulin in APS-SA series (*Q*_D_ = 500 mL/min, n = 3).

**Figure 8 membranes-12-00684-f008:**
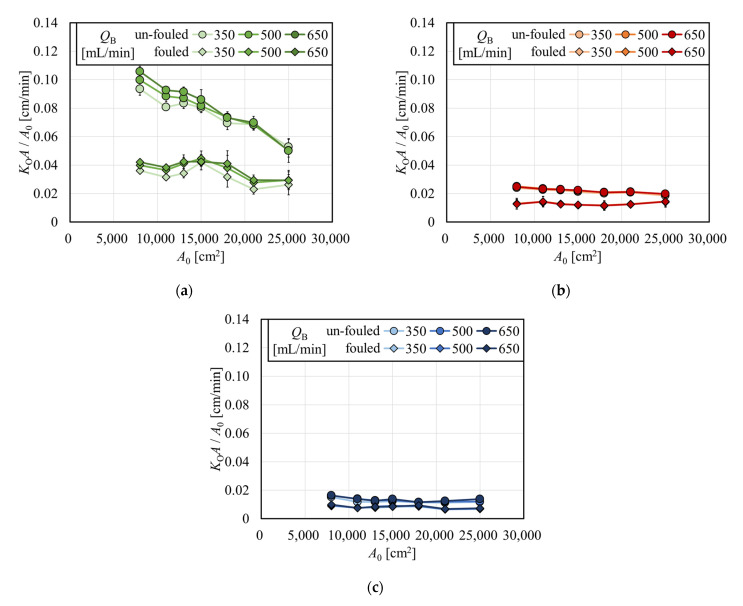
Overall mass transfer-area coefficient divided by *A*_0_ (*K*_O_*A*/*A*_0_) for (**a**) creatinine, (**b**) vitamin B_12_, and (**c**) inulin.

**Figure 9 membranes-12-00684-f009:**
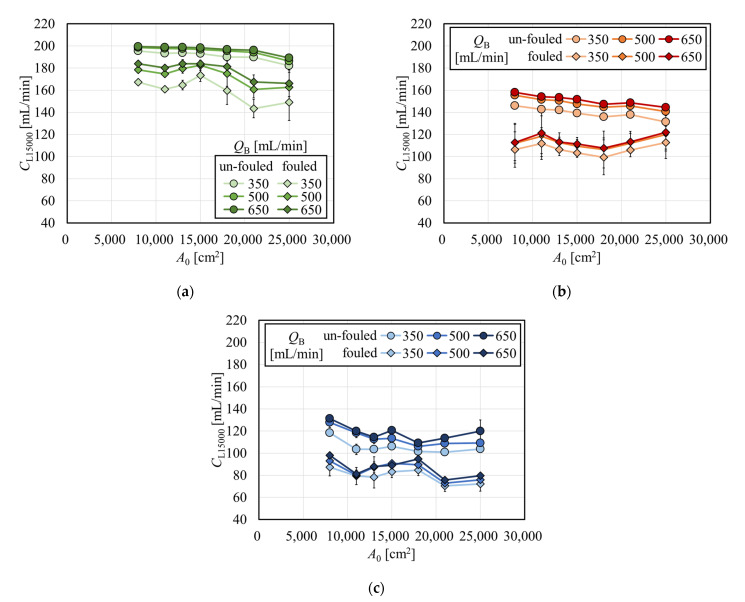
Clearances normalized to the effective surface area of 15,000 cm^2^ (*C*_L15000_) as a function of *A*_0_ for (**a**) creatinine, (**b**) vitamin B_12_, and (**c**) inulin.

**Table 1 membranes-12-00684-t001:** Technical specifications of APS-SA series.

				APS			
	08SA	11SA	13SA	15SA	18SA	21SA	25SA
Effective length, *L* [cm]	27.0	27.0	27.0	27.0	27.0	27.0	27.0
Arithmetic surface area, *A*_0_ [cm^2^]	8000	11,000	13,000	15,000	18,000	21,000	25,000
Inner diameter of fiber, *d* [µm]	183.5	183.5	183.5	183.5	183.5	183.5	183.5
Membrane thickness, Δ*x* [µm]	50	50	50	50	50	50	50
Inner diameter of the case, *dd* [cm]	2.8	3.2	3.4	3.8	4.0	4.2	4.6
Number of hollow fiber, *n* [-]	5139	7066	8351	9636	11,563	13,490	16,060

**Table 2 membranes-12-00684-t002:** Technical specifications of MFX-S eco series.

	MFX			
	11S eco	15S eco	21S eco	25S eco
Effective length, *L* [cm]	22.5	25.5	28.8	30.0
Arithmetic surface area, *A*_0_ [cm^2^]	11,000	15,000	21,000	25,000
Inner diameter of fiber, *d* [µm]	200	200	200	200
Membrane thickness, Δ*x* [µm]	40	40	40	40
Inner diameter of the case, *dd* [cm]	3.2	3.5	3.9	4.2
Number of hollow fiber, *n* [-]	7781	9362	11,605	13,263

## Data Availability

Not applicable.

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
