# Peer review of "Effect of Membrane Surface Area on Solute Removal Performance of Dialyzers with Fouling"

_membranes, 2022, doi:10.3390/membranes12070684_

Round 1
Reviewer 1 Report
The manuscript investigated the effect of filtration membrane surface area on several process parameters in dialyzers. The experiments were performed with and without membrane fouling using immobilized protein molecules. Overall, this manuscript is well written, easy to understand and the measured parameters should be interesting and helpful to the community. There are several minor revisions recommended before publication.
1. Line 65, please add units to the molecular weights
2. In figure 1, please make a more detailed figure description illustrating the symbols and color codes adopted here, e.g., please clarify the difference between lines with varied colors. What are the circles drawn on the flow lines in fig 1c?
3. line 130, why the author added coloring agents in the solution containing solutes?
4. Line 205, how was the rate of decrease calculated with a unit of mg/cm2?
5. Figure 3, instead of “intact”, the author may consider using “un-fouled” or “not fouled” to describe the membrane without immobilized protein molecules. Figure 3c and 3d don’t have curves from fouled data.
6. Figure 5, the author may consider combine the four sub-figures into one single figure.
7. Figure 6, please discuss the differences between curve trends of APS-SA and MFX-S eco. Why there was a decrease at A0=15000 for APS-SA?
8. Line 300, “middle molecules” is not a accurate word to describe molecules of certain molecular weights.
Reviewer 2 Report
This is an interesting paper considering fouling effect during hemodialysis and hemofiltration with a in vitro study design.
1. Artificial kidney (AK) is not re-use in most of the developed country.
So how is the effect of fouling in a complete new AK?
(compared with a re-used AK with heat sterilization?)
Is there any reference to support the clinical importance of fouling?
2. In a fresh new AK, only 4 hours dialysis hours, is there any reference
considering the fouling within such a short time duration?
( How is the effect of fouling at the beginning 1 hour and the last hour?)
How much is the percentage of clearance influenced by the fouling in a normal 4 hour dialysis?
3. Albumin immobilization is achieved by chemicals (glutaaldehyde).
Can that be reproduced or resemble in any condition in a normal status during dialysis?
(ex: such as in patients with hyperlipidemia or hyperparaproteinemia?)
How to measuring that effect in a normal hemodialysis? (if there is a well believed tool or method to measuring it?)
4. The effect of fouling, could that be clotting and obstruction of hollow fibers microscopically?
So the transfer area is shinking, as well as the dialysis clearance?
Or is this fouling related to the increasing thickness of membrane within hollow fiber? or both?
Both effects mentioned above made particles harder to transport across the membrane.
5. How to explain the sudden drop of packing density in APS-SA at 15000 cm2 in Figure 4?
Is that the manufactory company doing it on purpose in order to avoid high packing density (which cause decrease of clearance)?
Is there any packing difference between two kinds of AK?
Why is the other brand of AK do not have a sudden drop in packing density?
6. The math equations are not within my specialty, I will suggest checked by a math expert
to see if there are all correct, especially in Figure 5. (because it is totally theoretical.)
7. Blood flow (Qb) used in clincal setting is usually 250-350ml/min. This may not be major point because this is a in vitro study. However, the authors may adjust Qb to values which is more representative of normal dialysis in the future studies.
In a nephrologist view, the above mentioned questions are the main concerning points after I read the manuscript when I tried to link the results into my clinical practice.
